

# Surface-based observations of cold-air outbreak clouds during the COMBLE field campaign

Zackary Mages[1], Pavlos Kollias[1,2], Zeen Zhu[2], Edward P. Luke[2]

[1]School of Marine and Atmospheric Sciences, Stony Brook, NY, USA
[2]Environmental and Climate Sciences Dept., Brookhaven National Laboratory, Upton, NY, USA

*Correspondence to*:

Zackary Mages

Email: zackary.mages@stonybrook.edu





## Abstract

Cold-air outbreaks (CAOs) are characterized by extreme air-sea energy exchanges and low-level convective clouds over large areas in the high latitude oceans. As such, CAOs are an important component of the Earth's climate system. The CAOs in the Marine Boundary Layer Experiment (COMBLE) deployment of the US Department of Energy Atmospheric Radiation Measurement (ARM) Mobile Facility (AMF) provided the first comprehensive view of CAOs using a suite of ground-based observations at the northern coast of Norway. Here, cloud and precipitation observations from 13 CAO cases during COMBLE are analysed. A vertical air motion retrieval technique is applied to the Ka-band ARM Zenith-pointing Radar (KAZR) observations. The CAO cumulus clouds are characterized by strong updrafts with magnitudes between $2 - 8$ m s$^{-1}$, vertical extents of $1 - 3$ km, and horizontal scales of $0.25 - 3$ km. A strong relationship between our vertical air velocity retrievals and liquid water path (LWP) measurements is found. The LWP measurements exceed 1 kg m$^{-2}$ in strong updraft areas, and the vertical extent of the updraft correlates well with the LWP values. The CAO cumulus clouds exhibit large values of eddy dissipation rate. Finally, evidence of secondary ice production in the CAO cumulus clouds is presented.



## 1. Introduction

The Arctic is experiencing warming at the surface and throughout the troposphere at a rate faster than the
rest of the world, a phenomenon known as Arctic amplification (AA; Previdi et al., 2021). The warming
signal correlates in time and space with areas of significant sea-ice loss (Dai et al., 2019). Representing
AA in climate simulations requires a comprehensive understanding of the different climate feedbacks and
their impact on Arctic amplification, which include the Planck response and changes in water vapor plus
temperature lapse rate, surface albedo, and clouds (Forster et al., 2021). Not surprisingly, the cloud
feedback is particularly challenging to quantify (Zelinka et al., 2020). The latest Intergovernmental Panel
on Climate Change (IPCC) assessment report found low confidence that the Arctic cloud feedback is
positive, and it may even be slightly negative (Forster et al., 2021). Using reanalysis and satellite data,
Zhang et al. (2018) indicated a large uncertainty in the sign of the Arctic cloud feedback. An improved
understanding of high-latitude cloud systems, especially those over the open ocean such as cold-air
outbreaks (CAOs) is needed since AA is connected to greater sea-ice loss.

CAOs occur when cold, dry air is transported over the relatively warmer ocean, where the ocean surface
can then release large amounts of heat and moisture into the air (Pithan et al., 2018). Climatological
studies have highlighted the frequency at which CAOs occur in the Northern Hemisphere (Kolstad et al.,
2009; Kolstad, 2011; Fletcher et al., 2016a; Fletcher et al., 2016b; Smith and Sheridan, 2020), while others
have focused on the Arctic region specifically. Papritz and Spengler (2017) found that a high frequency
of CAOs occur in the Irminger and Nordic Seas while McCoy et al. (2017) found that December through
February is the season of maximum occurrence for open mesoscale cellular clouds, typical of CAOs, in
the Norwegian Sea. Brümmer and Pohlmann (2000) also found that, across ten winters, organized
convection in CAO events occurs more than 50% of the time over the Greenland and Barents Sea regions.
Most of these analyses are limited to reanalysis and satellite datasets, and more observational work is
crucial to our understanding of CAOs.





Early observational analyses of CAOs have focused on aircraft and sounding data from various field
campaigns around the globe (Lau and Lau, 1984; Hein and Brown, 1988; Chou and Ferguson, 1991;
Brümmer, 1996; Brümmer, 1997; Brümmer, 1999; Renfrew and Moore, 1999). Recently, work has been
done on data from the ACTIVATE (Aerosol Cloud meTeorology Interactions oVer the western ATlantic
Experiment) and ACCACIA (Aerosol-Cloud Coupling And Climate Interactions in the Arctic) field
campaigns that managed to capture some CAO events (Young et al., 2016; Seethala et al., 2021; Turnow
et al., 2021), and the MPACE (Mixed-Phase Arctic Cloud Experiment) field campaign, which provided
opportunity for ground-based observations of CAO events in Alaska (Shupe et al., 2008). Despite the
importance of CAO clouds, high resolution dynamical and microphysical observations, especially from
surface-based remote sensing facilities in the regions of Greenland and the Norwegian Sea where models
exhibit large inconsistencies, are not available (Pithan et al., 2014; Tomassini et al., 2017).

Here, analysis of surface-based observations from the Cold-Air Outbreaks in the Marine Boundary Layer
Experiment (COMBLE) field campaign are presented. Using profiling Doppler cloud radar, lidar, and
surface sensors, CAO events are identified, and the dynamical and microphysical properties of the shallow
convective clouds during CAOs are described. Section 2 describes the COMBLE field campaign and the
data used in this study. Section 3 describes the various data analysis methodologies used, including the
retrievals of vertical air motion, eddy dissipation rate, and the detection of secondary ice production.
Finally, we present our results in Section 4 and our conclusions in Section 5.

## 2. Data

From 1 December 2019 to 31 May 2020, the U.S. Department of Energy deployed the Atmospheric
Radiation Measurement (ARM) first mobile facility (AMF1) near the Norwegian Sea for the Cold-Air
Outbreaks in the Marine Boundary Layer Experiment (COMBLE) field campaign. The AMF1 was
located on the northern coast of Scandinavia at a latitude of 69.141° N, a longitude of 15.684° E, and an
altitude of 2 m above sea level while a smaller set of instruments were deployed at a latitude of 75° N on
Bear Island. The main objective of the experiment was to quantify the properties of shallow convective





clouds that develop as part of an air-mass transformation process when cold air advects over open water (Geerts et al., 2021). The two sites were located south of the Arctic ice edge, and the instruments successfully gathered comprehensive measurements of atmospheric conditions, clouds, precipitation, and aerosol that are used in this study.


The main instrument used in this study is the Ka-band ARM Zenith-pointing Radar (KAZR, Kollias et al., 2016), a zenith-pointing Doppler cloud radar operating at 35 GHz. The KAZR has a vertical resolution of 30 m and a temporal resolution of 2 s (Kollias et al., 2020). In this study, the radar reflectivity factor and mean Doppler velocity are used. For polarimetry, we supplement our dataset with Doppler spectra observations from the collocated Ka-band ARM Scanning Cloud Radar (Ka-SACR, Kollias et al., 2014a) during times of vertically pointing operation. The Ka-SACR has a vertical resolution of 49.96 m and a temporal resolution of 2.97 s. Liquid water path (LWP) estimates were provided by a microwave radiometer (MWR) that operates at 23.8 GHz and 31.4 GHz to determine column-integrated water vapor and liquid water along the vertical line-of-sight path (Morris, 2019). The balloon-borne sounding system (SONDE) and the Interpolated Sonde (INTERPSONDE) value-added product are used to retrieve profiles of atmospheric conditions over AMF1 (Holdridge, 2020; Toto and Jensen, 2016). The eddy dissipation rate (EDR; Borque et al., 2016) retrievals use the horizontal wind variables from INTERPSONDE as inputs, and the updraft width (chord length) calculations use the horizontal wind variables from SONDE. For the cloud base height (CBH) data, we use a ceilometer which sends a laser pulse at a 910 nm wavelength to detect light scattered by clouds and precipitation (Morris, 2016). Finally, to understand the type of precipitation reaching the surface at AMF1, we use a PARSIVEL2 laser disdrometer (Bartholomew, 2020). The PARSIVEL 1-min particle size distributions and fall velocities were used to characterize the precipitation type and intensity.

### 3. Methodology


3.1 CAO events selection



During the COMBLE field campaign, 34 CAO days were identified over the 6-month period for a total
of 19% of the campaign (Geerts et al., 2022). Here, data from 13 of these days are used. All periods with
prefrontal and frontal cloud systems are removed, and our analysis focuses on the dynamical and
microphysical characteristics of the periods with shallow convective CAO clouds. The cases and time
periods are listed in Table 1.

### 3.2 Vertical Air Motion Retrieval


The KAZR mean Doppler velocity ($V_D$) contains contributions from the vertical air motion ($V_{AIR}$) and
the reflectivity-weighted particle sedimentation velocity ($V_{SED}$). The estimation of $V_{AIR}$ requires the
removal of $V_{SED}$ from the observed $V_D$ (Kollias et al., 2022; Zhu et al., 2021). Well-established techniques
applicable to profiling Doppler cloud radar observations (Protat and Williams, 2011; Kalesse and Kollias,
2013) exist that can provide adaptive relationships between $V_{SED}$ and radar reflectivity factor (Z). These
relationships have been evaluated in stratiform precipitation systems where the $V_{AIR}$ is weak, and its mean
value is near zero averaged over a 20 minute or longer period. Lamer et al. (2015) indicates that such
relationships are challenging to develop in cumulus clouds due to the preferential presence of strong
updraft motions. The presence of updrafts will bias low (reduce) the hydrometeor size distribution $V_{SED}$
and can, in many cases, result in positive (upward) hydrometeor motion. The preliminary perusal of the
KAZR and MWR CAO observations indicated that strong updraft motions ($V_D > 2$ m s$^{-1}$) are usually
found in vertical in-cloud atmospheric columns. These updraft occurrences clearly correlate well with
periods when the MWR detects the presence of columns with liquid water exceeding 0.25 kg m$^{-2}$. Here,
we use the apparent relationship between LWP and $V_{AIR}$ by filtering out all KAZR observations when the
LWP exceeded 0.25 kg m$^{-2}$. All other KAZR observations are used to estimate the relationship between
$V_{SED}$ and Z using the methodology proposed by Protat and Williams (2011). To confirm our choice of
LWP threshold, we found the relationship between $V_{SED}$ and Z below cloud base for all 13 cases in both
the high (> 0.25 kg m$^{-2}$) and low (< 0.25 kg m$^{-2}$) LWP periods. The relationships were very similar (not
shown), meaning similar types of particles are falling below cloud base in both regions.




First, the KAZR signal-to-noise ratio (SNR) values are used to identify the locations of hydrometeors using the Hildebrand and Sekhon (1974) threshold technique (Kollias et al., 2014). The KAZR observations during periods with LWP < 0.25 kg m$^{-2}$ are sorted into reflectivity bins with widths of 1.5 dB in the range of -20 to +20 dBZ. In each reflectivity bin, the corresponding $V_D$ are used to estimate the median Doppler velocity. The median Doppler velocity is our best estimate of the $V_{SED,BE}$ for the radar reflectivity values within a particular bin. Reflectivity bins that contain less than 1% of the KAZR observations are discarded due to their small sample size. The bin pairs of radar reflectivity and corresponding $V_{SED,BE}$ create a look-up table (no fit is attempted), and this process is repeated for each CAO case. Figure 1a shows the Doppler velocity box plots for each reflectivity bin on 28 March; the other 12 CAO cases (not shown) exhibit a similar behaviour. Despite this, we do not create a global Z-$V_D$ look-up table; rather, each day's data is used to capture the smaller difference unique to the day. Sensitivity tests for these fits are performed using two other LWP thresholds of 0.5 kg m$^{-2}$ and 0.8 kg m$^{-2}$, and the differences are negligible. The $V_{SED,BE}$ values are between $0.5 - 2$ m s$^{-1}$, which is consistent with the presence of frozen hydrometeors. The relationship between the $V_{SED,BE}$ and KAZR radar reflectivity indicates a gradual increase in the sedimentation velocity with radar reflectivity. The joint distribution of the KAZR radar reflectivity and $V_D$ for all 13 CAO cases is shown in Fig. 1b. The KAZR radar reflectivity shows a broad distribution with most echoes between ±20 dBZ. The KAZR $V_D$ distribution is centred at 0.5 - 0.75 m s$^{-1}$. For each CAO event, the relationship between radar reflectivity and $V_{SED,BE}$ is used to estimate the vertical air motion $V_{AIR}$ using the following expression:

$$V_{AIR}(Z[t,h]) = V_D(Z[t,h]) - V_{SED,BE}(Z[t,h])$$

where t is time and h is height (range) of the KAZR observations. The uncertainty in the $V_{AIR}$ estimates is controlled by the uncertainty of the $V_{SED,BE}$ estimates since the primary measurement ($V_D$) has negligible uncertainty (below 0.1 m s$^{-1}$). The uncertainty of the $V_{SED,BE}$ is controlled by the number of samples used in the estimation of the median $V_D$ value within each radar reflectivity bin and the variability of the cloud microphysics during the sampling period. The $V_{SED,BE}$ estimates for the same radar reflectivity bin differed very little from one CAO case to another (< 0.25 kg m$^{-2}$). The same uncertainty





was found when shorter time periods were used. Using a fairly conservative approach, the uncertainty of
the $V_{SED,BE}$ is between $0.3 - 0.4$ m s$^{-1}$.

For a subset of the observational period, we analyse Ka-SACR Doppler spectra for evidence of secondary
ice production. $V_{AIR}$ for this analysis is estimated using the slower falling edge of the spectral energy
density's principal peak as a proxy for $V_{AIR}$, including a downward adjustment of $0.28$ m s$^{-1}$ to compensate
for turbulence broadening. For our analysis, we avoid strong downward $V_{AIR}$ values, as this is when the
presence of a viable air motion tracer particle population is most uncertain. We thus enforce a lower limit
of -1 m s$^{-1}$ on $V_{AIR}$. Our primary measurements in this analysis are linear depolarization ratio (LDR)
determined by the ratio of crosspolarized to copolarized spectral energy density, calibrated copolarized
spectral reflectivity normalized to units of dBZ s m$^{-1}$, and spectral terminal fall speed computed as the
difference between $V_{AIR}$ and spectral $V_D$.

### 3.3 Eddy Dissipation Rate Retrieval

EDR is retrieved from the KAZR Doppler velocity and the INTERPSONDE sounding product using the
algorithm outlined by Borque et al. (2016). The INTERPSONDE sounding product is first interpolated to
2 s – 30 m resolution to be consistent with the KAZR observations. Next, at each height, 20 minutes of
the Doppler velocity time series is selected to generate the corresponding velocity power spectrum ( S(*f*)
) by performing an FFT. Assuming the turbulence to be homogeneous, in the region of the inertial
subrange, EDR can be estimated as:


$$\varepsilon = \frac{2\pi}{V} \left\{ \frac{2}{3\alpha} \int_{f_1}^{f_2} S(f)df \right\} (f_1^{-1.5} - f_2^{-1.5})^{-1.5}$$

where $\varepsilon$ is the retrieved EDR, V is the horizontal wind obtained from the sounding product, $\alpha$ is the
Kolmogorov constant and is taken as 0.5, and $f_1 - f_2$ is the lower and upper frequency limit in the inertial
subrange. It is apparent that the accuracy of the $\varepsilon$ retrieval is highly dependent on the selection of the




inertial subrange, which is determined by the frequency interval ($f_1 - f_2$). Here, we adapt the same approach proposed by Borque et al. (2016) to confine the inertial subrange: 33 frequency pairs are predefined for each selected frequency interval, a power law fitting is performed for the velocity spectrum S($f$) and only the fitting slopes within -5/3 ± 1/3 are selected as "good inertial subranges" and used for $\varepsilon$

estimation. Finally, the retrieved $\varepsilon$ from all the "good intervals" are averaged to obtain the $\varepsilon$ product.

### 3.4 Updraft Dimensions

The $V_{AIR}$ retrievals are used to estimate properties of coherent updrafts in CAOs. A conditional threshold

of $V_{AIR} > 2$ m s$^{-1}$ is used to identify spatially coherent updraft structures. The 2 m s$^{-1}$ conditional velocity threshold is much higher than the $V_{AIR}$ uncertainty (0.3 – 0.4 m s$^{-1}$). This will ensure that we detect the presence of an updraft. In addition, the 2 m s$^{-1}$ $V_{AIR}$ threshold is the typical vertical air motion value used to distinguish stratiform versus convective conditions. Using the algorithm outlined in Kollias et al., (2014b), a low-pass filter that is five vertical profiles wide and five range gates deep is applied to the $V_{AIR}$

estimates three times. This allows for the identification of coherent updraft structures.

The coherent updraft structures are then analysed to estimate their depth (vertical extent), width (chord length), and range of magnitudes within the structure. The distance between the lowest and highest KAZR range gates that a coherent updraft occupies is used to estimate their vertical extent. A similar approach

is used to estimate the temporal duration of the updraft structures. This duration value is then multiplied by the average horizontal wind speed in the lowest 5 km of the atmosphere from the nearest sounding in time to give the chord length of the updraft. Finally, the range of magnitudes is given by all the values within the same coherent updraft structure. A sensitivity analysis was performed to evaluate the impact of the selected threshold for determining coherent updraft structures. Using a conditional threshold of

$V_{AIR} > 1$ m s$^{-1}$, we found negligible differences in the updraft statistics. Finally, due to a lack of soundings, 1 December only contributes to updraft vertical extent and magnitude results.





## 4. Results

An example three-hour period of CAO observations from 28 March 2020 is shown in Fig. 2. Several

cumulus clouds were detected by the KAZR with cloud tops between 3.5 to 4.5 km. The surface

temperature averaged 0.76 °C for the period and never dropped below 0 °C; meanwhile, temperatures

ranged from -44.6 °C to -37.5 °C near cloud top. Throughout the cloud layer, the lapse rate was about 8.4

°C km$^{-1}$, and the prevailing wind was predominantly from the northwest with at most 8 – 9° of wind shear.

In COMBLE, the KAZR was operated only in co-polar mode. The lack of KAZR linear depolarization

ratio observations prevents us from reliably using radar Doppler spectra techniques for a hydrometeor

phase classification (Kalesse et al., 2016; Luke et al., 2021). No melting layer signature is detected in the

KAZR radar reflectivity and mean Doppler velocity observations (Fig. 2b) throughout the observing

period. The KAZR reflectivity exceeds +25 dBZ in the shallow convective cores, indicating the presence

of large hydrometeors or a high number concentration. The retrieved $V_{AIR}$ is shown in Fig 2c. Several

deep updraft structures are observed within the same shallow convective cloud suggesting the presence

of boundary layer organization. In particular, the cumulus cloud detected around 10 UTC exhibits four

distinct updrafts with $V_{AIR}$ values higher than 5 m s$^{-1}$. Similar coherent updraft structures are commonly

observed throughout the COMBLE dataset. On the other hand, there are cumulus clouds with negligible

or no updraft structures; this is a result of sampling clouds at different stages of their evolution. The MWR

detected the presence of significant LWP (exceeding 1 kg m$^{-2}$) in the areas where updrafts were retrieved.

The collocation of the updraft structures with the presence of supercooled liquid provided confidence in

the $V_{AIR}$ estimates. Finally, the shallow convective clouds exhibit high EDR estimates reaching values up

to 0.01 m$^2$ s$^{-3}$. Similarly with the presence of updrafts, the EDR values are higher in the areas of active

shallow convective clouds (Fig. 2d).


4.1 Updraft Structure Analysis.

One of the main scientific drivers of the COMBLE field campaign is to better understand mixed-phase

cloud processes and improve their representation in high resolution numerical models (Geerts et al.,





2022). In-cloud updrafts are very important in microphysics. The observed distribution of updraft chord

length from all CAO cases is shown in Fig. 3a. The distribution peaks at updraft chord lengths less than

500 m, and more than 80% of the observed updrafts have chord lengths less than 1 km. Similarly, the

observed distribution of updraft vertical extent peaks at a value less than 500 m. About 5% of the observed

updrafts have vertical extents higher than 1 km. The distribution of the range of updraft magnitudes (Fig.

260  3c) shows that most of the updrafts are defined by $V_{AIR}$ values between 2 and 3 m s$^{-1}$, but some have

values as high as 8-9 m s$^{-1}$. These values far exceed the ones found by Brümmer (1999) in aircraft data

from the ARKTIS field campaign over the same region.

The observed scales of CAO cumulus updrafts are unresolved by current Global Cloud Resolving Models

265  (Satoh et al., 2019). Since the KAZR data has such a high temporal resolution, we transform the KAZR

time-height data to horizontal distance-height data using the mean horizontal wind speed from the lowest

5 km of the atmosphere from the nearest soundings in time. This allows us to look at the updraft chord

length and magnitude at different horizontal resolutions: 250 m in Figures 4a and 4c and 1 km in Figures

4b and 4d. We take the median KAZR profile over each distance interval, and we run our low-pass filter

270  only once. As the horizontal resolution becomes coarser, KAZR identifies fewer updrafts, and they are

losing their impressive magnitudes. We also attempted a 3 km resolution, but none of the updrafts were

resolved. The 250 m resolution distributions still closely resemble those seen in Fig. 3, so increasing a

model's resolution beyond 250 m will hinder its ability to resolve the structures in CAOs.

275  We also examine the updraft magnitude profile as a function of normalized updraft depth. The observed

updraft structures are classified into three categories based on their vertical extent: those with depths less

than 1 km (Fig. 5a), those with depths between 1 and 2 km (Fig. 5b), and those with depths greater than

2 km (Fig. 5c). In general, the deeper updrafts are associated with stronger vertical air motions.

Throughout the normalized updraft depths, 55% of $V_{AIR}$ values are greater than 3 m s$^{-1}$ in Fig. 5a, 66%

of $V_{AIR}$ values are greater than 3 m s$^{-1}$ in Fig. 5b, and 75% of $V_{AIR}$ values are greater than 3 m s$^{-1}$ in Fig.

5c. Finally, the whiskers in Fig. 5c show the deepest updrafts have both the strongest and weakest $V_{AIR}$

values.


### 4.2 EDR Analysis


The distribution of the EDR measurements as a function of height above the surface for all CAO cases is shown in Fig. 6. The highest EDR values ($10^{-3} - 10^{-1}$ $m^2$ $s^{-3}$) are observed near the surface. This is consistent with the strong surface forcing that characterizes CAO cloud systems. At higher altitudes, the EDR distribution is broader. There is a region where EDR steadily decreases with height and another

region where EDR consistently stays above a value of $10^{-3}$ $m^2$ $s^{-3}$. Given that stratus clouds typically have a range of EDR values between $10^{-5}$ and $10^{-3}$ $m^2$ $s^{-3}$, some of these CAO cumuli display strong turbulence throughout their depth while the strong turbulence is concentrated closer to the surface for others.

### 4.3 Relationship between LWP and updrafts


In Figure 2, the presence of liquid water is well-correlated with the coherent updraft structures. This relationship is further investigated using all the COMBLE observations (Fig. 7). In general, the LWP correlates well with the square of the depth of the cloud (Wood, 2012; Fan et al., 2018). In Figure 7a, the measured LWP is plotted against the sum of $V_{AIR}$ values in the updraft physical depth. Similarly, the LWP

in the column is plotted against the maximum $V_{AIR}$ value in the updraft physical depth. These two relationships exhibit a plausible agreement between two independent measurements, which further supports the good performance of the $V_{AIR}$ retrieval technique. In Figure 7c, we plot histograms of vertical air velocity values in atmospheric columns with differing LWP values. When the LWP measurement is higher than 0.25 kg $m^{-2}$ for our cases, the atmospheric column has larger values of $V_{AIR}$, both upward and

downward, as indicated by the large amount of data in the tails of the histogram compared to the histogram with smaller LWP values. When KAZR identifies upward motion, the MWR measures a large amount of liquid, but those occurrences are sparse. Figure 7d shows nearly 75% of the LWP data is near zero, while only about 1% of the LWP data is higher than 2 kg $m^{-2}$.

### 4.4 Hydrometeor fraction profile




Here, the relationship between updrafts and hydrometeor fraction profile is investigated. First, the KAZR data are separated into hour-long periods, and the hydrometeor fraction is calculated at each range gate. The hydrometeor fraction is the fraction of KAZR significant meteorological detections over the total number of KAZR profiles during the one-hour period. In addition, the updraft fraction is estimated as the fraction of retrieved $V_{AIR} > 2$ m s$^{-1}$ over the total number of KAZR significant meteorological detections during the one-hour period. The distribution of the hourly-estimated hydrometeor fraction as a function of normalized cloud height is shown in Fig. 8a. The maximum hydrometeor fraction gradually reduces towards the cloud top. The observed hydrometeor fraction profile suggests surface and cloud base conditions determine the overall cloud fraction in CAOs. No evidence of hydrometeor detrainment near the cloud top is observed, as it has been observed in the shallow oceanic convection (Lamer et al., 2015). The dataset is further classified into three CAO cloud thickness types: cloud top heights (CTHs) less than 3.5 km, CTHs between 3.5 and 4.5 km, and CTHs greater than 4.5 km (Fig. 8b). Despite their considerable differences in CTH, the hydrometeor fraction estimates near the cloud base are clustered around 0.52 – 0.6.

The updraft fraction profiles increase towards the cloud top (Fig. 8b). This is a combination of the updraft structures being vertically oriented and the overall hydrometeor fraction reduction with height above the cloud base. The mean updraft fraction near the cloud base for the three different CAO cloud top cases exhibit higher co-variability with CTH. The updraft fraction at the cloud base more than doubles between the shallow (CTH < 3.5 km) and the deep (CTH > 4.5 km) CAO cases. This further suggests that near cloud base, conditions are important for determining the vertical extent of the CAO cumulus field.

### 4.5 Secondary Ice Production

The presence of strong updrafts and high supercooled liquid amounts within the temperature range of -3 to -8 °C suggests the possibility for secondary ice production (SIP) within the CAO cumulus clouds. Luke et al. (2021) presents a comprehensive observational study that utilizes polarimetric radar Doppler spectra





to detect and quantify the occurrence of SIP. The KAZR did not collect polarimetric observations during
COMBLE, but the collocated Ka-band Scanning ARM Cloud Radar (Ka-SACR; Kollias et al., 2014)
spent time vertically pointing as part of its nominal sampling pattern. Using polarimetric radar Doppler
spectra recorded by the Ka-SACR on 31 December 2019, we apply the method of Luke et al. (2021) to
detect and quantify the occurrence of secondary ice production in the temperature range of -3 to -8 °C.
On that day, this temperature range extends from 500 to 1000 m in altitude (Fig. 9b). We detect the
presence of Doppler spectra bin observations dominated by a columnar ice crystal habit by those having
an LDR between -16 and -14 dB. We then aggregate these bins according to their terminal fall speed and
divide their quantity by the total number of bins with a measurable LDR, aggregated in the same way by
terminal fall speed. We require the copolarized and crosspolarized spectral energy density to both be at
least 4 dB above their noise floor for LDR to be measurable. We then know the fraction of hydrometeors
that can be attributed to a columnar ice crystal habit as a function of terminal fall speed. Figure 9c shows
that this fraction is enhanced at the altitude corresponding to a temperature of -5 °C in the fall speed range
of small needles. For comparison, we compute the fraction of Doppler spectra bins dominated by spherical
hydrometeors as above by subsetting the numerator to observations having an LDR between -22 and -20
dB. As seen in Figure 9d, minimal enhancement occurs near -5 °C. Finally, following Luke et al. (2021),
we determine the secondary ice multiplication factor of needle detections using the baseline detection
threshold of -21 dBZ s m$^{-1}$, which is shown in Fig. 9e. Occurrences of secondary ice multiplications from
10x to 100x are readily apparent, with additional occurrences in the range of 100x to 1000x.
Unfortunately, the Ka-SACR operations in a vertically pointing mode were not regularly executed, thus
limiting our ability to conduct a comprehensive study of SIP detection and occurrence. Nevertheless, the
one case analysed clearly indicates the presence of SIP in CAOs.

## 5. Conclusions

The COMBLE observations provide the first systematic long-term, cloud-scale, ground-based remote
sensing dataset of Arctic CAOs. Observations from a profiling cloud radar (KAZR), a ceilometer and a
microwave radiometer (MWR) are used to study the cloud-scale dynamics of 13 CAO events. The KAZR





observations are used to estimate CAO cumulus cloud properties such as hydrometeor fraction, cloud top height, vertical air motion, and EDR. The LWP measurements from the MWR and their relationship to
cloud dynamics are investigated.

The CAO cellular shallow convective clouds observed at Andenes, Norway have typical cloud top heights between 3 and 5 km, and the average hydrometeor fraction is 50 – 60%. Owing to the large surface sensible heat flux, CAO cumulus clouds are characterized by strong updraft structures. The distribution
of retrieved updraft magnitudes peaks at 3 m s$^{-1}$, but a considerable number of updrafts have vertical air motion values that exceed 4 – 5 m s$^{-1}$. On the other hand, the coherent updraft structures have narrow widths that peak at 250 m and vertical extents typically around 500 m. Representing these updraft structures in numerical models requires high resolution modelling at the scales of 100 – 200 m horizontal spacing. The LWP time series indicates the intermittent presence of liquid columns with LWP values in
excess of 1 – 2 kg m$^{-2}$. Furthermore, the intermittent spikes in LWP amount correlate with the detection of coherent updraft structures and their vertical extent. The EDR retrieval confirms the turbulent nature of the CAO cumulus clouds with the highest values near cloud base ($\sim 5 \times 10^{-3}$ m$^2$ s$^{-3}$).

The CAO cumulus hydrometeor fraction profile peaks at the cloud base level (0.5 – 0.6) and gradually
decreases with height above the cloud base. The cloud base hydrometeor fraction profile exhibits little relationship to the cumulus field cloud top height. On the other hand, the cumulus field cloud top height exhibits better covariance with the updraft fraction profile.

In addition, we show that secondary ice production is present during a cold-air outbreak with ice
multiplication factors approaching three orders of magnitude. This is consistent with a growing body of evidence suggesting that updraft regions containing supercooled liquid are favourable for secondary ice production.

The presented work provides valuable information for model intercomparison studies that will attempt to
understand mixed-phase cloud processes in CAOs, but there is a limitation. Our work examines the cumuli





from a Eulerian perspective and is restricted to a two-dimensional view of the atmosphere; we cannot speak to the evolution of the clouds nor to their three-dimensional geometry and organization. Future work may begin by looking at data collected by the Norwegian weather radar network and Ka-SACR during COMBLE, where one can analyse the mesoscale organization of the cumuli and their three-

dimensional structure as they evolve in time.

*Data Availability*: The ARM observational datasets for COMBLE are available at the ARM Data Centre. The KAZR data (kazrge) can be accessed via http://dx.doi.org/10.5439/1498936. The Ka-SACR data (kasacrcfrvpt) can be accessed via http://dx.doi.org/10.5439/1482713. The ceilometer data (ceil) can be

accessed via http://dx.doi.org/10.5439/1181954. The microwave radiometer data (mwrlos) can be accessed via http://dx.doi.org/10.5439/1046211. The sounding data (sondewnpn) can be accessed via http://dx.doi.org/10.5439/1595321. The interpolated sounding data (interpolatedsonde) can be accessed via http://dx.doi.org/10.5439/1095316. The PARSIVEL data can be accessed via http://dx.doi.org/10.5439/1779709. The rest of the COMBLE observational datasets can be accessed via

https://adc.arm.gov/discovery/#/results/iopShortName::amf2019comble.

*Author Contributions*: Z.M. and P.K. designed the methodology, and Z.M. performed the analysis. Z.Z. performed the EDR retrieval, and Z.Z. and E.L. prepared the data. Z.M. and P.K. prepared the manuscript, with contributions from Z.Z. and E.L.


*Competing Interests*: The authors declare that they have no conflicts of interest.

*Acknowledgements*: We would like to acknowledge the data support provided by the Atmospheric Radiation Measurement (ARM) Program sponsored by the U.S. Department of Energy.


*Financial Support*: The contributions of P.K. and E.L. were supported by contract DE-SC0012704 with the U.S. Department of Energy (DOE).





## Appendix A

The PARSIVEL2 disdrometer provides 1-min observations of hydrometeor size and fall velocity (Fig. A1a). The disdrometer observations have been successfully used in previous studies to classify the types (i.e. phase, density, size) of hydrometeors reaching the surface. Here, the PARSIVEL2-based hydrometeor identification developed by Friedrich et al. (2013a) and Friedrich et al. (2013b) and the size-velocity fits for unrimed particles in Locatelli and Hobbs (1974) are used to assess the hydrometeor typing

in CAOs (Fig. A1b). Throughout the 13 cases, the rain hydrometeor type dominates, as it accounts for about 58% of the total particles detected by the disdrometer (Fig. A1a). Hail, graupel, and snow only account for 0.52%, 3.51%, and 0.06% of the total particles detected, respectively. The PARSIVEL2-based hydrometeor classification contradicts the KAZR observations. First, no noticeable attenuation is observed in the KAZR observations. Furthermore, the distribution of KAZR Doppler velocities at 300 m

above the surface indicates that most of the values in the distribution are around 1.5 m s$^{-1}$ (Fig. A1c). The KAZR Doppler velocity measurements were compared against those recorded by the Ka-SACR. The comparison shows excellent agreement between the two radars. The discrepancy between the KAZR and disdrometer observations suggests the possibility of i) artifacts in the disdrometer observations due to the orientation of the disdrometer and/or the near-surface wind magnitude and/or ii) the presence of numerous

irregularly-shaped particles that are difficult to characterize using the disdrometer.





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



| Day | Hours (Local Time) |
|---|---|
| December 1st, 2019 | 0:00-2:09; 20:18-24:00 |
| December 2nd, 2019 | 0:00-15:24 |
| December 31st, 2019 | 0:00-18:15 |
| January 4th, 2020 | 9:00-24:00 |
| January 21st, 2020 | 2:42-17:48 |
| January 22nd, 2020 | 4:42-16:00 |
| February 2nd, 2020 | 4:43-22:09 |
| February 5th, 2020 | 0:00-24:00 |
| March 13th, 2020 | 1:46-4:01; 8:12-24:00 |
| March 27th, 2020 | 0:00-4:36; 10:39-14:13 |
| March 28th, 2020 | 4:09-24:00 |
| March 29th, 2020 | 0:00-24:00 |
| April 10th, 2020 | 1:00-1:48; 7:04-15:04 |

**Table 1. Dates and times used from the COMBLE field experiment.**





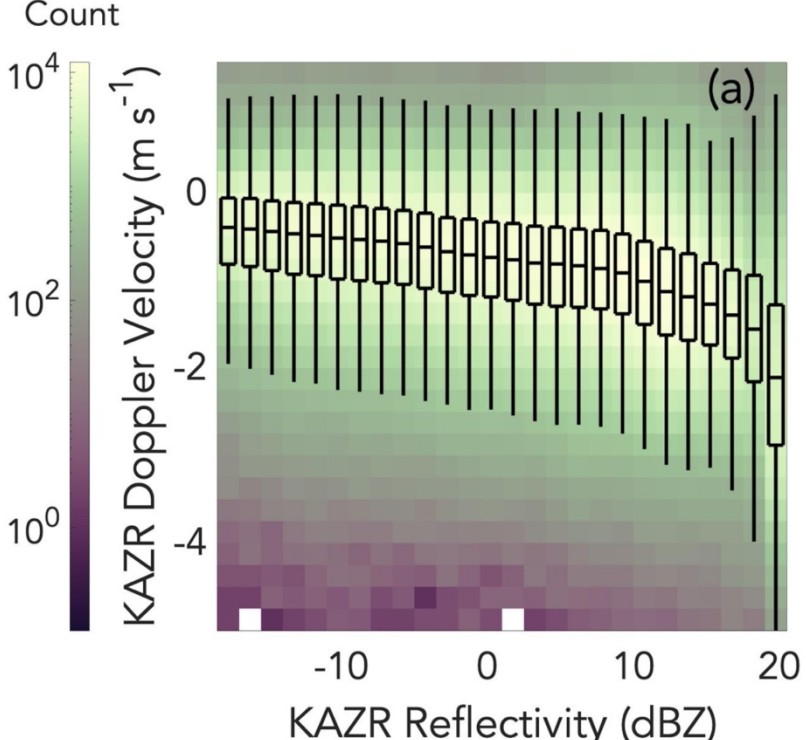

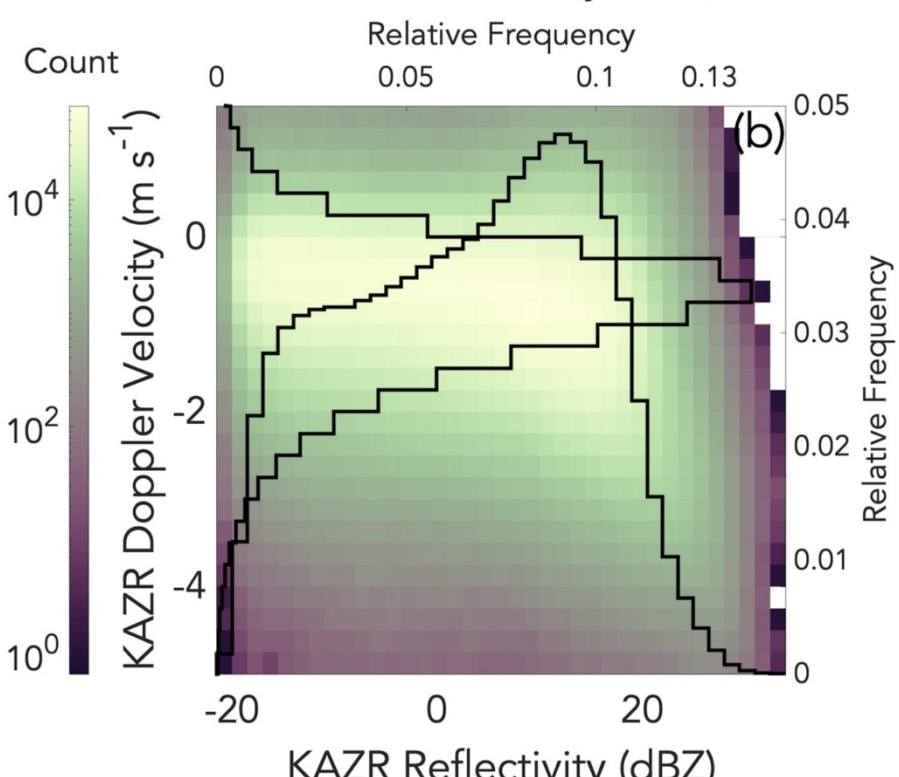





**Figure 1. (a) For March 28th, 2020, a joint PDF of KAZR reflectivity and Doppler velocity in the vertical profiles with a liquid water path (LWP) value less than 0.25 kg m⁻² and box-and-whisker plots showing the median Doppler velocity in every 1.5 dB reflectivity bin; (b) For all 13 COMBLE cases, a joint PDF of KAZR reflectivity and Doppler velocity in the vertical profiles with a LWP value less than 0.25 kg m⁻², the relative frequency distribution of KAZR Doppler velocity in the same low LWP periods, and the relative frequency distribution of KAZR reflectivity in the same low LWP periods.**







**Figure 2. Time-height mapping from 8-11 LT on March 28th, 2020 of a) KAZR radar reflectivity, b) KAZR Doppler velocity, c) retrieved vertical air velocity and liquid water path, and (d) retrieved eddy dissipation rate.**





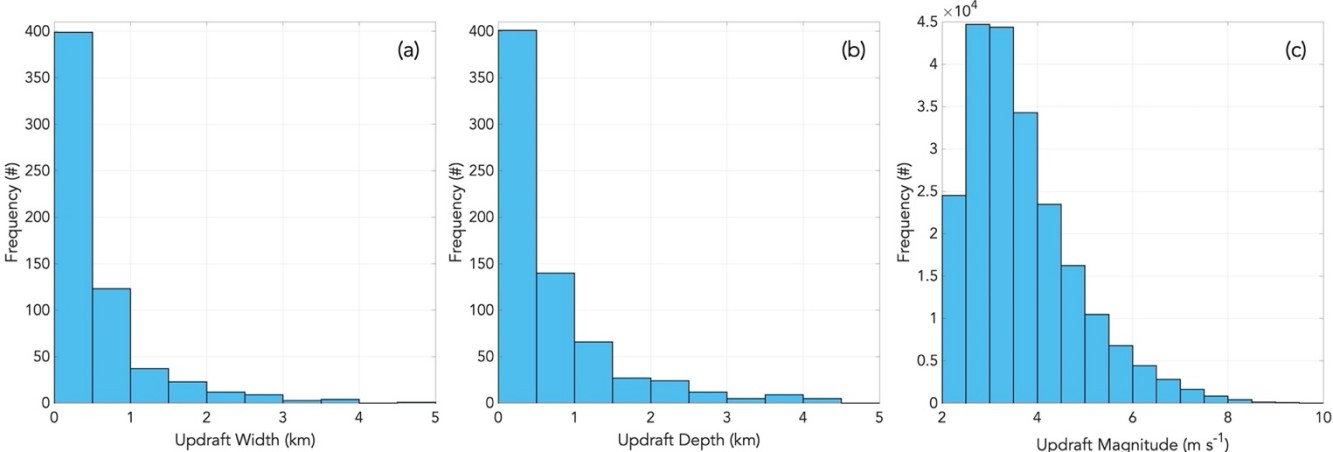

**Figure 3. For all 13 COMBLE cases, histograms of a) updraft chord length (width) in km, b) updraft vertical extent in km, and c) the range of magnitudes in the updraft in ms⁻¹.**


**Figure 4. From all 13 COMBLE cases, histograms of updraft chord length (width) in km and range of magnitudes in the updraft in ms$^{-1}$ at a horizontal resolution of (a-b) 250 m and (b-d) 1 km.**





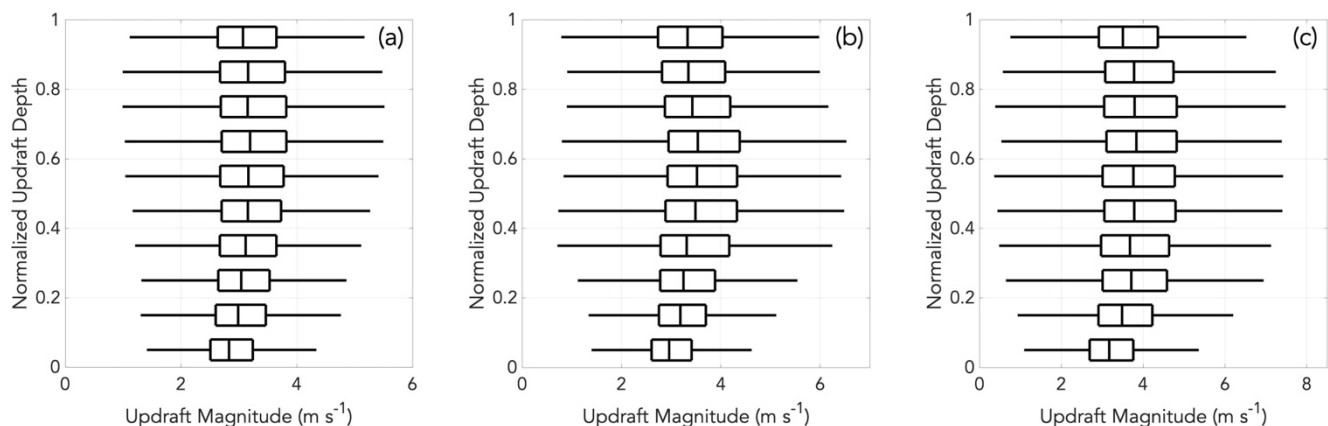


**Figure 5. For all 13 COMBLE cases, box-and-whisker plots of updraft magnitude as a function of normalized updraft depth for (a) updrafts with a depth less than 1 km, (b) updrafts with a depth between 1 and 2 km, and (c) updrafts with a depth greater than 2 km.**







**Figure 6. For all 13 COMBLE cases, a joint PDF of eddy dissipation rate versus height.**







**Figure 7.** For all 13 COMBLE cases, (a) the mean and standard deviation of the sum of vertical air velocity ($V_{AIR}$) values in the updraft depth for each liquid water path (LWP) bin of width 0.25 or 0.5 km m$^{-2}$, b) the mean and standard deviation of the maximum $V_{AIR}$ value in the updraft depth for each LWP bin of width 0.25 or 0.5 kg m$^{-2}$, (c) histograms of $V_{AIR}$ values in the atmospheric column when LWP is greater than 0.25 kg m$^{-2}$ (red) and when LWP is less than 0.25 kg m$^{-2}$ (blue), and (d) the relative frequency of LWP values in bins of width 0.1 kg m$^{-2}$.






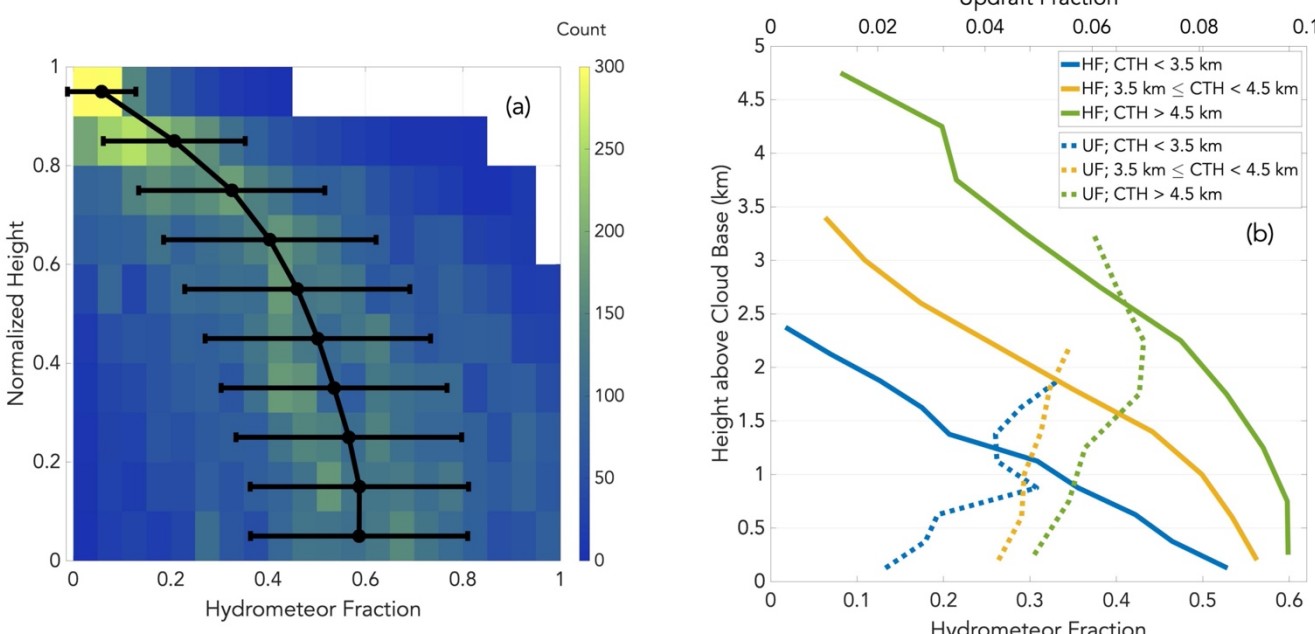

**Figure 8. For all 13 COMBLE cases, (a) a joint pdf of hydrometeor fraction (HF) versus normalized**
**height, along with a mean profile of HF as a function of normalized height and (b) mean profiles of**
**HF (black) and updraft fraction (UF; blue) as a function of height above cloud base, with solid lines**
**for cloud tops less than 3.5 km, dashed lines for cloud tops between 3.5 and 4.5 km, and dotted lines**
**for cloud tops above 4.5 km.**





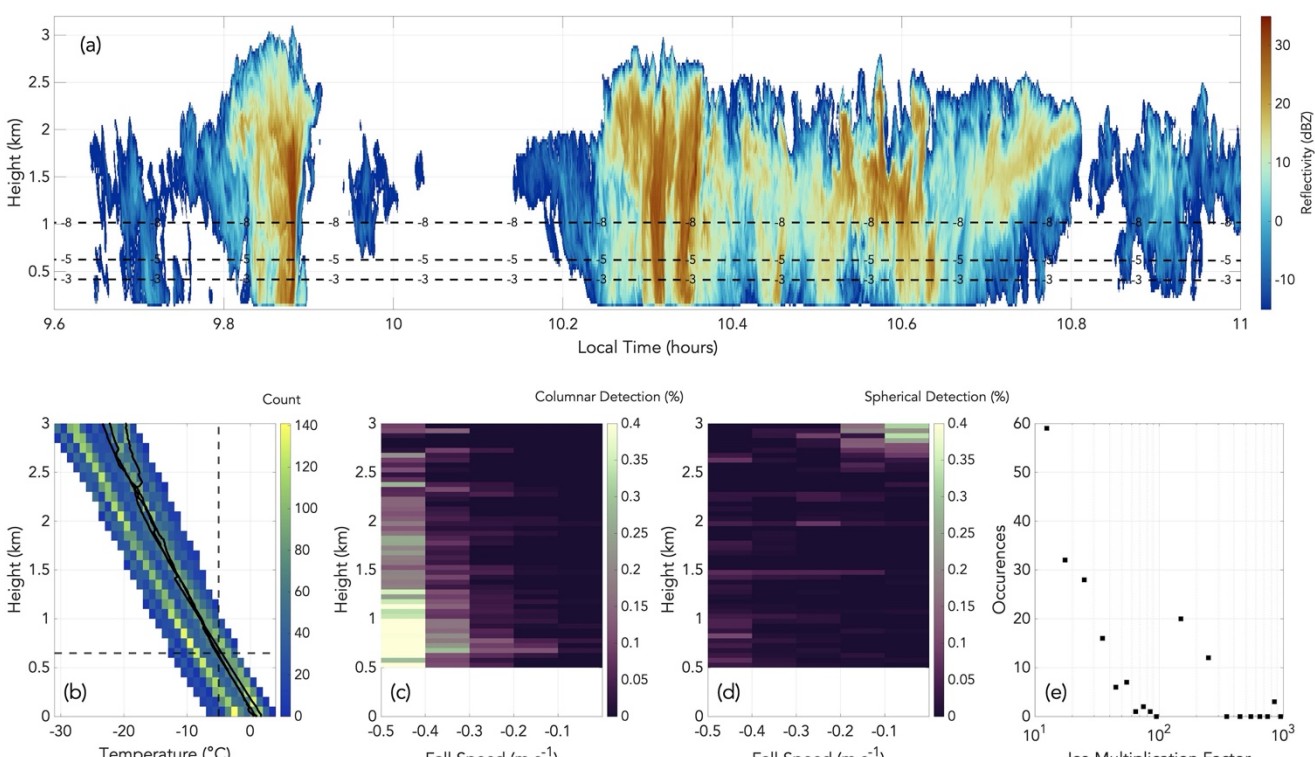

**Figure 9. For December 31st, 2019, (a) time-height mapping during 9:36-11 LT of KAZR radar reflectivity (colors) and INTERPSONDE isotherms (dashed black lines); (b) a joint PDF of temperature and height (colors) from 32 radiosondes launched during the 13 cases along with the three temperature profiles from December 31st (black); and the percentage of Doppler spectra with columnar detections (c) and spherical detections (d) in each fall speed-height bin during 0-18:15 LT.**

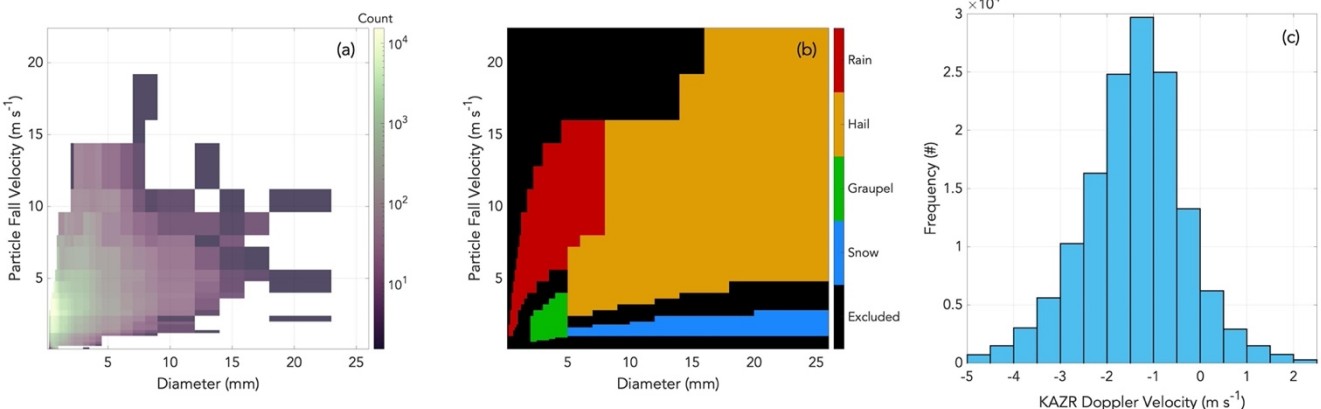



**Figure A1. For all 13 COMBLE cases, (a) a joint PDF of disdrometer particle diameter and particle fall velocity, (b) a hydrometeor identification map used to categorize the precipitation type at the surface, and (c) a histogram of KAZR Doppler velocities at 300 m.**