# Peer review of "Surface-based observations of cold-air outbreak clouds during the COMBLE field campaign"

_Atmospheric Chemistry and Physics, 2022_

## Referee Comment (RC1)

Reviewer comments to authors of "Surface-based observation of cold-air outbreak clouds during the COMBLE field campaign."

In this paper, the authors characterized low-level cumulus clouds during the Cold-air Outbreak (CAO) in Marine Boundary Layer Experiment (COMBLE) using ground-based observations. They investigated 13 COMBLE cases of low-level convective clouds, and found a general understanding of cloud dynamical properties (e.g., vertical air motion and eddy dissipation rate) related to thermodynamical quantities (e.g., liquid water path). Last, the authors presented the presence of secondary ice production in one available case (31 December 2019). The characteristics of these low convective clouds are important to improve model parameterizations, and response to a changing climate. This scientific information will be very useful to many scientific and stakeholder communities.

I have three serious concerns about the manuscript, in addition to specific questions and comments listed as below.

1. The thresholds in this study are somewhat arbitrary and the lack of background. The review needs to understand why the authors choose those thresholds. This is important because the results should be changed in how the authors selected the thresholds. I strongly recommend that the authors should add an exact explanation (or specific background) in choosing the below thresholds.

- (1) Line 119: 13 cases why did the authors choose those cases?
- (2) Line 120: Why are the prefrontal and frontal clouds neglected?

(3) Line 135–143: LWP threshold of 0.25 - I cannot find a strong relationship between KAZR observation and LWP > 0.25 kg m-2. Also, the frequencies are too small when you choose LWP < 0.25 kg m-2 as a low LWP period in Figure 7. Please add the percentage of LWP data as the author mentioned in Line 307–308.

(4) Line 268: the horizontal resolution of 250 m and 1 km. Why did the authors choose the resolution of 250 m? Also, the reviewer does not convince the data conversion from time-height to horizontal distance-height. Since the KAZR is a vertically pointing radar, this data is unable to explain (or represent) the horizontal distribution associated with the model resolution.

(5) Line 277: three categories of updraft depths (1 km and 2 km). The results should be changed when the authors choose different categories.

(6) Line 323: cloud thickness types - cloud top heights (CTH) of 3.5 km and 4.5 km. I can not find any results and references why the authors choose the CTH of 3.5 km and 4.5 km to categorize the cloud thickness.

2. Overall, the author presented the results without detailed physical interpretation. I don't want to point them out here. Please add a more detailed physical interpretation in the result section.

3. Introduction: the reviewer suggests adding more research background in the introduction.

(1) Please review the previous studies using the COMBLE field campaign

(2) There are some recent field campaigns the authors mentioned in Line 64–71. What are the differences compared to previous field campaigns?

(3) Previous field campaign (i.e., ACTIVATE) collected the observational data for the high-resolution dynamic and microphysical observation.

Minor comments:

1. Line 139–140: Please add the figure if the authors want to explain the relationship.

2. Line 150: What is the "VSED,BE"?

3. Line 170: I do not find how authors can calculate the uncertainty (below 0.1 m s-1).

4. Line 190: What do intervals for the sonde observation? I assume the authors collected the sonde data every 6 hours, then interpolated 2 sec. Can this interpolated data (2 sec) compare with KAZR? The reviewer thinks that this interpolated data for horizontal winds cannot correlate with updrafts derived from KAZR due to large time differences. Can you estimate the uncertainty of eddy dissipation rates? If so, please add the uncertainties.

5. Line 241: Please add a local time

6. Line 245: There are peaks of LWP (> 0.25 kg m-2) around 8.3–8.5 hours and ~10.8 hours in Figure 2c. What is the reason for those peaks?

7. Line 251: Remove "."

8. Line 287:  $10^{-1}$  m2 s-3, I do not think this value is correct, because there is no frequency between log10-1.7 to log10-1.5. Please recheck this value.

9. Line 288: What is "the strong surface forcing"? Does it mean "surface sensible heat flux"?
10. Line 290: Please add reference about "above a value of 10-3 m2 s-3"

11. Line 291: "strong turbulence" – Please add a correct explanation.

12. Line 300–301: Does it correct? How can the authors argue the correlation (or relationship) with  $R^2$ =0.121, 0.153?

12. Line 299: Suggest removing "physical"

13. Line 318: Please add the meaning of the normalized height at 1 or 0. I assume normalized height at 1 would be cloud top height, right?

14. Line 378: Suggest changing "modeling"

---

## Author Response (AR1)

Reviewer comments to authors of "Surface-based observation of cold-air outbreak clouds during the COMBLE field campaign."

In this paper, the authors characterized low-level cumulus clouds during the Cold-air Outbreak (CAO) in Marine Boundary Layer Experiment (COMBLE) using ground-based observations. They investigated 13 COMBLE cases of low-level convective clouds and found a general understanding of cloud dynamical properties (e.g., vertical air motion and eddy dissipation rate) related to thermodynamical quantities (e.g., liquid water path). Last, the authors presented the presence of secondary ice production in one available case (31 December 2019). The characteristics of these low convective clouds are important to improve model parameterizations, and response to a changing climate. This scientific information will be very useful to many scientific and stakeholder communities.

Thank you to Reviewer #1 for providing valuable feedback. It is much appreciated, and it improved our paper!

I have three serious concerns about the manuscript, in addition to specific questions and comments listed as below:

**Comment 1**: The thresholds in this study are somewhat arbitrary and the lack of background. The review needs to understand why the authors choose those thresholds. This is important because the results should be changed in how the authors selected the thresholds. I strongly recommend that the authors should add an exact explanation (or specific background) in choosing the below thresholds.

**Response**: This is a very good general comment. Our group has extensive experience in the application of cloud radar observations for cloud and precipitation microphysics, and we hope that our explanations will provide sufficient information to the reviewer and to the readers of the manuscript. One challenge in our analysis is that CAOs have not been previously observed by an ARM-like observing facility for such an extensive period, and to a certain extent, we had to come up with some criteria/thresholds that are based on the available observations.

**Comment 1a**: In Line 119, why did the authors choose those 13 cases?

**Response**: In the recently published BAMS article that describes the COMBLE field experiment (The COMBLE Campaign: A Study of Marine Boundary Layer Clouds in Arctic Cold-Air Outbreaks, Bulletin of the AMS, Geerts et al. 2022, doi:10.1175/BAMS-D-21-0044.1), a total of 39 CAO cases were identified based on the boundary layer thermodynamic structure. We decided to focus on 13 of them using the following criteria: The selected cases were long-lived (several hours, while many of the identified cases in the BAMS article were very short-lived), and the cloud field morphology was relatively simple (i.e., avoid cases with extensive upper-level cloud presence that interacted with the underlying cumulus field that was the primary focus of our study).

**Comment 1b**: In Line 120, why are the prefrontal and frontal clouds neglected?

**Response**: We wanted to focus on the convective cumulus that develop from the strong surface heat fluxes in the cold air outbreak system. Any period with just prefrontal/frontal clouds does not fall into the regime we want to study, and any period with both prefrontal/frontal clouds and convective cumulus may have competing effects or be too complex.

**Comment 1c**: In Lines 135-143, the LWP threshold of 0.25 - I cannot find a strong relationship between KAZR observation and LWP > 0.25 kg m$^{-2}$. Also, the frequencies are too small when you choose LWP < 0.25 kg m$^{-2}$ as a low LWP period in Figure 7. Please add the percentage of LWP data as the author mentioned in Line 307–308.

**Response**: Another excellent comment. The strong relationship that we are referring to is that of upward motions (positive KAZR Doppler velocities) and LWP values from the MWR. This correlation is easy to see when perusing time series data like those illustrated in Fig. 2b. In Figure 7, we tried to illustrate this relationship, but the original version of Figure 7 does not capture this relationship. Thus, we have revised one of the panels of Figure 7, and we now show the relationship between the LWP measurements from the MWR and the maximum observed KAZR Doppler velocity in the column. We will adjust Figure 7 and make the appropriate changes to Section 4.3.

**Change**: Visual inspection of Figure 2 suggests a correlation between the presence of liquid water and the coherent updraft structures. This relationship is further investigated using all the COMBLE observations (Fig. 7). Looking at the LWP measurements broadly, Figure 7a shows nearly 75% of the LWP data is near zero, while only about 1% of the LWP data is higher than 2 kg m$^{-2}$. Meanwhile, Figure 7b shows that as LWP in the column increases, so too does the maximum $V_D$, which reinforces the visual inspections we made and our use of LWP as a threshold for our vertical air motion retrieval. In general, the LWP correlates well with the square of the depth of the cloud (Wood, 2012; Fan et al., 2018). In Figure 7c, the measured LWP is plotted against the sum of $V_{AIR}$ values in the updraft depth. Similarly, in Figure 7d, the LWP in the column is plotted against the maximum $V_{AIR}$ value in the updraft depth. These two relationships exhibit a plausible agreement between two independent measurements, which further supports the good performance of the $V_{AIR}$ retrieval technique.

[Figure]

**Figure 7. For all 13 COMBLE cases, (a) the relative frequency of liquid water path (LWP) in bins of width 0.1 kg m$^{-2}$, (b) the median, 25$^{th}$, and 75$^{th}$ percentile of the maximum Doppler velocity (V$_D$) in the atmospheric column for each LWP bin of width 0.25 kg m$^{-2}$ or 0.5 kg m$^{-2}$, (c) the median, 25$^{th}$, and 75$^{th}$ percentile of the sum of vertical air velocity (V$_{AIR}$) in the updraft depth for each LWP bin of width 0.25 or 0.5 kg m$^{-2}$, and (d) the median, 25$^{th}$, and 75$^{th}$ percentile of the maximum V$_{AIR}$ in the updraft depth for each LWP bin of width 0.25 or 0.5 kg m$^{-2}$.**

**Comment 1d**: In Line 268, with regards to the horizontal resolution of 250 m and 1 km, why did the authors choose the resolution of 250 m? Also, the reviewer does not convince the data conversion from time-height to horizontal distance-height. Since the KAZR is a vertically pointing radar, this data is unable to explain (or represent) the horizontal distribution associated with the model resolution.

**Response**: As part of the COMBLE CAOs investigation, a model intercomparison study is underway. The simulations are being set up with sub-kilometer grid spacing (250-500 m), while other regional model runs will be conducted at kilometer-scale resolution. Thus, our aim was to adapt our study to provide them useful information. At resolutions better than 250 m, our results closely resembled the original KAZR measurements too closely.

**Comment 1e**: In Line 277, with regards to the three categories of updraft depths (1 km and 2 km), the results should be changed when the authors choose different categories.

**Response**: We did not use literature to create these groups. We categorized the data in a way that gave 3 subgroups of nearly equal size. We will make note of this. We were also limited by the data itself, where cloud tops peaked at 5 km or less. This limited the amount of depth classes we could choose. Finally, we chose thresholds that were easily reproducible with models.

**Change**: The observed updraft structures are classified into three categories of nearly equal size based on their vertical extent: those with depths less than 1 km (Fig. 5a)…

**Comment 1f**: In Line 323, with regards to the cloud thickness types (CTH of 3.5 km and 4.5 km), I cannot find any results and references as to why the authors choose the CTH of 3.5 km and 4.5 km to categorize the cloud thickness.

**Response**: We did not use literature to create these groups. We categorized the data in a way that gave 3 subgroups of nearly equal size. We will make note of this. Similarly to Comment 1e, we chose thresholds that were easily reproducible with models.

**Change**: The dataset is further classified into three CAO cloud thicknesses types by splitting it into 3 samples of nearly equal size: cloud top heights (CTHs) less than 3.5 km…

**Comment 2**: Overall, the author presented the results without detailed physical interpretation. I don't want to point them out here. Please add a more detailed physical interpretation in the result section.

**Response**: The reviewer is correct; we decided to describe our results and avoid physical interpretation. There are couple of reasons for this. First, we did not want to overreach by adding physical interpretation based on incomplete data/information. For example, we do know that CAOs occur under strong air-sea interaction and flux conditions, but these surface conditions are not available in COMBLE. In addition, our observations are limited to profiling, and the role of mesoscale organization will be examined using modeling in follow-up studies. Finally, our analysis identified and quantified CAO cumulus dynamics for the first time, and we feel that a detailed description of our dynamical retrievals is sufficient. We are currently working on calibrating the KAZR observations and developing a hydrometeor classification that along with a complete analysis of the secondary ice production climatology in these cloud systems will form the basis of a follow-up study that will be more integrated and will contain more physical interpretation.

**Comment 3**: The reviewer suggests adding more research background in the introduction.

**Comment 3a**: Please review the previous studies using the COMBLE field campaign.

**Response**: To our knowledge, only one scientific paper has been published using the COMBLE field campaign.

**Change**: Here, analysis of surface-based observations from the Cold-Air Outbreaks in the Marine Boundary Layer Experiment (COMBLE) field campaign are presented. Initial work has been done using satellite data on two COMBLE cases (Wu and Ovchinnikov, 2022), and this study will focus also focus on measurements taken during the campaign. Using profiling Doppler cloud radar…

**Citations Added**:
Wu, P., and Ovchinnikov, M.: Cloud Morphology Evolution in Arctic Cold-Air Outbreak: Two Cases During COMBLE Period, Journal of Geophysical Research: Atmospheres, 127, https://doi.org/10.1029/2021JD035966, 2022.

**Comment 3b**: There are some recent field campaigns the authors mentioned in Lines 64–71. What are the differences compared to previous field campaigns?

**Response**: Thank you for this comment. We need to be more precise with our wording.

**Change**: Early observational analyses of CAOs have focused on aircraft and sounding data from various field campaigns around the globe (Lau and Lau, 1984; Hein and Brown, 1988; Chou and Ferguson, 1991; Brümmer, 1996; Brümmer, 1997; Brümmer, 1999; Renfrew and Moore, 1999). Recently, work has been done on data from the ACTIVATE (Aerosol Cloud meTeorology Interactions oVer the western ATlantic Experiment) and ACCACIA (Aerosol-Cloud Coupling And Climate Interactions in the Arctic) field campaigns that managed to capture some CAO events (Young et al., 2016; Seethala et al., 2021; Turnow et al., 2021), although studying CAOs was not the main goal of the campaigns. The MPACE (Mixed-Phase Arctic Cloud Experiment) field campaign also provided opportunity for ground-based observations of CAO events in Alaska (Shupe et al., 2008). However, there are other regions in the Northern Hemisphere where ground-based observations of CAOs are lacking. Despite the importance of CAO clouds, high resolution dynamical and microphysical observations, especially from surface-based remote sensing facilities in the regions of Greenland and the Norwegian Sea where models exhibit large inconsistencies, are not available (Pithan et al., 2014; Tomassini et al., 2017).

**Comment 3c**: Previous field campaigns (i.e., ACTIVATE) collected the observational data for the high-resolution dynamic and microphysical observations.

**Response**: This is correct. However, to our knowledge, the ACTIVATE campaign did not sample CAOs, at least with the characteristics that we observed during COMBLE in the Norwegian Sea in the Arctic.

Minor comments:

**Comment 4**: In Lines 139-140, please add the figure if the authors want to explain the relationship.

**Response**: We have added this to Figure 7. See above in response to Comment 1c.

**Comment 5**: In Line 150, what is the "$V_{SED,BE}$"?

**Response**: BE stands for Best Estimate. We will make note of this.

**Change**: The median Doppler velocity is our best estimate (BE) of the $V_{SED,BE}$ for the radar…

**Comment 6**: In Line 170, I do not find how authors can calculate the uncertainty (below 0.1 m s⁻¹).

**Response**: The uncertainty of the KAZR Doppler velocity measurement can be estimated using well-established relationships (Doviak and Zrnic, 1984).

$$var(V_d) = \frac{\lambda \cdot PRF^2}{2 \cdot M} \left[ \frac{\sigma_{vn}}{4\sqrt{\pi}} + 2 \cdot (\sigma_{vn})^2 \cdot \frac{N}{S} + \frac{1}{12}\left(\frac{N}{S}\right)^2 \right]$$

where $\sigma_{vn}$ is the normalized spectrum width

$$\sigma_{vn} = \frac{\sigma_v}{2 \cdot V_{Nyquist}}$$

and N/S is the ratio of noise to signal (linear units), $V_{Nyquist}$ is the Nyquist velocity of the radar and $\sigma_v$ is the spectrum width. For typical COMBLE values of $\sigma_v = 0.3$ ms⁻¹, $V_N = 10.75$ ms⁻¹, M = 10000, and N/S = 0.1, the standard deviation of the KAZR Doppler velocity is 0.15 ms⁻¹. We will include this formula in the paper, and we will replace 0.1 with 0.15 ms⁻¹.

**Change**: …has negligible uncertainty (below 0.15 m s⁻¹). The uncertainty of the …

**Comment 7**: In Line 190, what are the intervals for the sonde observations? I assume the authors collected the sonde data every 6 hours, then interpolated 2 sec. Can this interpolated data (2 sec) compare with KAZR? The reviewer thinks that this interpolated data for horizontal winds cannot correlate with updrafts derived from KAZR due to large time differences. Can you estimate the uncertainty of eddy dissipation rates? If so, please add the uncertainties.

**Response**: ARM's INTERPSONDE data product provides high time-resolution profiles of atmospheric state variables like temperature and relative humidity and is widely used and cited. We do not create the interpolated sounding grid on our own and instead use this.

**Comment 8**: In Line 241, please add a local time.

**Response**: This comment made us aware of an inconsistency in the time we use across the paper. Everything should be in UTC, so we will correct the x-axis and caption of Figure 2 as well as the

column title for Table 1. We will make note of the time in both UTC and local time (UTC+1) as you request.

**Change**: In particular, the cumulus cloud detected around 10 UTC/11 LT exhibits four…

[Figure]

**Comment 9**: In Line 245, there are peaks of LWP (> 0.25 kg m⁻²) around 8.3–8.5 hours and ~10.8 hours in Figure 2c. What is the reason for those peaks?

**Response**: The peaks coincide with an active cumulus passing over AMF1. These peaks are the response of the microwave radiometer to the microwave emissions of liquid, here specifically supercooled liquid. These regions of supercooled liquid are coincident with the updrafts. We

posit that other KAZR echoes that pass without peaks in LWP are dissipating cumulus that have reached glaciation.

**Comment 10**: In Line 251, remove ".".

**Response**: We accept this change.

**Comment 11**: In Line 287, I do not think $10^{-1}$ $m^2$ $s^{-3}$ is correct, because there is no frequency between $\log_{10}(-1.7)$ to $\log_{10}(-1.5)$. Please recheck this value.

**Response**: You are right; $10^{-1}$ $m^2$ $s^{-3}$ is not correct. It should be $10^{-2}$ $m^2$ $s^{-3}$. Thank you!

**Change**: The highest EDR values ($10^{-3}$ $m^2$ $s^{-3}$ – $10^{-2}$ $m^2$ $s^{-3}$) are observed near the surface.

**Comment 12**: In Line 288, what is "the strong surface forcing"? Does it mean "surface sensible heat flux"?

**Response**: Yes, we mean "strong surface sensible heat flux". We will change our wording.

**Change**: This is consistent with strong surface sensible heat fluxes that characterize CAO cloud systems.

**Comment 13**: In Line 290, please add reference about "above a value of $10^{-3}$ $m^2$ $s^{-3}$".

**Comment 14**: In Line 291, please add a correct explanation for "strong turbulence".

**Response**: We will make changes to address Comments 13 and 14 at the same time.

**Change**: Two modes appear, one where the EDR steadily decreases with height and another where EDR stays constant with height. Overall, the strongest turbulence in the distribution is concentrated in the lowest 2 km between values of $10^{-3}$ and $10^{-2}$ $m^2$ $s^{-3}$. The two stratiform EDR profiles shown in Borque et al. (2016) do not share these characteristics. One does not have as a deep a layer as shown here, and the other has values hovering around $10^4$ $m^2$ $s^{-3}$, one order of magnitude less than shown here.

**Comment 15**: In Lines 300-301, does it correct? How can the authors argue the correlation (or relationship) with $R^2 = 0.121, 0.153$?

**Response**: In the new Figure 7, we do not include correlations anymore. See the response to Comment 1c.

**Comment 16**: In Line 299, suggest removing "physical".

**Response**: We accept this change.

**Comment 17**: In Line 318, please add the meaning of the normalized height at 1 or 0. I assume normalized height at 1 would be cloud top height, right?

**Response**: We will add an explanation.

**Change**: The distribution of the hourly-estimated hydrometeor fraction as a function of normalized cloud height, where 0 represents cloud base and 1 represents cloud top, is shown in Fig. 8a.

**Comment 18**: In Line 378, suggest changing "modeling".

**Response**: We accept this change.

This is the review of the manuscript entitled "Surface-based observations of cold-air outbreak clouds during the COMBLE field campaign" written by Mages et al.

Generally, I found the study quite interesting and scientifically worth to publish. However, in quite many passages, the authors stay vague with their statements. They leave open questions, which need to be resolved before one can actually trust the argumentation line of the authors. Respective passages in the text hint to deficiencies in the applied techniques, which either need to be addressed, discussed, or negated. That's the reason why I had to provide a rather long list (of partly short) major concerns. I need to see detailed answers in order to be convinced that the presented results and conclusions are indeed reasonable and defendable. I thus strongly recommend a second review round.

Thank you to Reviewer #2 for providing valuable feedback. It is much appreciated, and it improved our paper!

Major concerns:

**Comment 1**: In Line 137, please prove the statement "These updraft occurrences CLEARLY CORRELATE WELL with periods when the MWR detects the presence of columns with liquid water exceeding 0.25 kg m$^{-2}$." using statistical methods. Or do you mean that the visual inspection of the time series suggests a correlation?

**Response**: An excellent comment. We agree that the use of the expression "clearly correlate well" is unfortunate. The relationship we are referring to is that of upward motions (positive KAZR Doppler velocities) and LWP values from the MWR. This correlation is easy to see when perusing time series data like those illustrated in Figure 2c. In Figure 7, we tried to illustrate this relationship, but the original version of Figure 7 does not capture this relationship. Thus, we have revised one of the panels of Figure 7, and we now show the relationship between the LWP measurements from the MWR and the maximum observed KAZR Doppler velocity in the column.

**Change**: A preliminary visual inspection of the KAZR and MWR CAO observations indicated that strong updraft motions indicated by the positive KAZR Doppler velocity measurements ($V_D$ > 2 m s$^{-1}$) are usually during periods when the MWR indicated the presence of high values of liquid water path. A LWP threshold of 0.25 kg m$^{-2}$ was selected to identify these periods. The selected periods have very little sensitivity to the selected LWP threshold.

**Comment 2**: In Lines 142-144, wouldn't also the Doppler lidar data (doi: 10.5439/1178583) help to validate the retrieval? One could use the Doppler-lidar-derived vertical velocity at cloud base to obtain the motion of the high number of small cloud droplets, which is most-likely the air motion. These values could then be correlated to the Doppler velocity and reflectivity observations of KAZR.

**Response**: This is a very insightful comment, and we could not agree more with this suggestion. We were surprised at the beginning phase of this study when we learned that the Doppler lidar was not collocated with the profiling radar. Clearly a missed opportunity because as the reviewer

suggests, the coincident measurements from the two instruments would have helped with the subsequent analysis. We originally intended to do our vertical velocity retrievals with KAZR and the Doppler lidar in tandem. The Doppler lidar is located nearly 600 km to the north at an ancillary site on Bear Island while KAZR is located at AMF1 in Andenes.

**Comment 3**: In Line 156, why do different Z-$V_D$ relationships apply to each day? What are the potential meteorological drivers?

**Response**: Figure 9b shows how variable temperature profiles were during our cases, with two main regimes appearing. A varying temperature profile would influence our Z-$V_D$ relationships, and without looking at other meteorological variables, we decided to apply a daily Z-$V_D$ relationship to each case rather than a COMBLE-wide one. There are certainly enough nuances on each day to warrant more exact relationships; we did not want to lose any of the small dynamical features unique to each day.

**Comment 4**: You motivate why you need to remove values exceeding 0.25 kg m$^{-2}$ in lines 137-140. Here, in lines 157-158, you did the test also for higher LWP values and it also works? How and why?

**Response**: We tested different LWP thresholds to confirm our choice of 0.25 kg m$^{-2}$. For each threshold, we plotted the Z-$V_D$ relationships in the convective regions and the non-convective regions. For the 0.25 kg m$^{-2}$ threshold, both relationships had similar shapes, but for the other two thresholds, you could see evidence of the updrafts influencing the relationship, especially in the higher reflectivity bins. This confirmed that our 0.25 kg m$^{-2}$ isolated the convective regions. We will add a figure to the appendix to show this.

**Change**: Sensitivity tests for these fits are performed using two other LWP thresholds of 0.5 kg m$^{-2}$ and 0.8 kg m$^{-2}$, and we found that once we exceeded 0.25 kg m$^{-2}$, the convective updrafts began influencing the Z-$V_D$ relationships, supporting this threshold's ability to isolate the convection. Figures S1c and S1e show the median $V_D$ values in the higher reflectivity bins approaching and even exceeding zero. Also noteworthy is that the median $V_D$ values in Figures S1a and S1b are similar while the ones in Figures S1c and S1d and Figures S1e and S1f are not, reinforcing our choice of LWP threshold.

[Figure]

**Figure S1. For March 28ᵗʰ, joint PDFs of KAZR reflectivity and Doppler velocity in the vertical profiles with a liquid water path (LWP) greater than 0.25 kg m⁻² (a), less than 0.25 kg m⁻² (b), greater than 0.5 kg m⁻² (c), less than 0.5 kg m⁻² (d), greater than 0.8 kg m⁻² (e), and less than 0.8 kg m⁻² (f). Solid lines indicate the median Doppler velocity in each reflectivity bin for that LWP threshold, and dashed lines indicate the median Doppler velocity in each reflectivity bin for the other two LWP thresholds.**

**Comment 5**: In Lines 159-160, shouldn't also the observed particle phase state play an important role in the selection of valid datapoints? Ice/snow falls very differently and produces very different reflectivities, compared to liquid water droplets.

**Response**: We agree. However, no liquid-dominated radar signatures are observed in the analyzed dataset. No melting layer signature is observed, and the Doppler velocities and their relationship to the observed radar reflectivity points to solid hydrometeors.

**Comment 6**: In Line 177, can you provide a literature basis for the retrieval? It is similar to what was done by Li et al. in the group of D. Moissev? (Li, H., Möhler, O., Petäjä, T., and Moisseev, D.: Two- year statistics of columnar-ice production in stratiform clouds over Hyytiälä, Finland: environmental conditions and the relevance to secondary ice production, Atmos. Chem. Phys., 21, 14671–14686, https://doi.org/10.5194/acp-21-14671-2021, 2021.)

**Response**: We want to be sure that the reviewer understands that our COMBLE secondary ice retrieval is based on the methodology of Luke et al. published in PNAS: Luke, E. P., Yang, F., Kollias, P., Vogelmann, A. M., and Maahn, M. New insights into ice multiplication using remote-sensing observations of slightly supercooled mixed-phase clouds in the Arctic. *Proceedings of the National Academy of Sciences* 118, e2021387118, doi:10.1073/pnas.2021387118 (2021). While the objectives of Luke et al. and Li et al. have similarities, they apply different methodologies. Probably the best way to specifically answer the reviewer's question is the following direct quote from Li et al.: "Our results are similar to the conclusion reached by Luke et al. (2021), who used a different approach for establishing the range of ice crystal concentration from radar observations." We appreciate the reviewer's question as we are pleased when other studies corroborate with our findings.

**Comment 7**: In Lines 178-179, where is the spectral energy density defined? You might just add this variable name to Line 100, where Ka-SACR is introduced and where you only write about 'Doppler spectra'.

**Comment 8**: In Lines 179-180, how comes this value of 0.28 m s$^{-1}$? Is turbulence always the same so that this correction is always the same? I would expect that different corrections are required, depending on the convective situation. How would a variable 'correction value' affect the overall retrieval?

**Response**: This response addresses both Comments 7 and 8. The reviewer is correct to point out that the description of the technique for the identification of SIP areas in CAOs in incomplete. We will revise.

**Change**: During COMBLE, there were periods when the Ka-band Scanning ARM Cloud Radar (Ka-SACR, Kollias et al., 2014a,b) was pointing vertically. During these periods, the Ka-SACR recorded co- and cross-polar radar Doppler spectra. The radar Doppler spectrum represents the frequency (velocity) distribution (spectral density, $mm^6\,m^{-3}$ / $ms^{-1}$) of the background radar signal at a particular range. In a vertically-pointing radar, the Doppler spectra provide the distribution of backscattered signal, and the backscattered signal's intensity is controlled by the hydrometeor's number concentration and size over a range of Doppler velocities. These

velocities are dependent on the hydrometeor's sedimentation velocity and the vertical air motion fluctuations within the radar sampling volume. The cross-polar Doppler spectrum provides information about the location (velocity) of non-spherical particles. The recorded co- and cross-polar radar Doppler spectra can be used as input to a novel retrieval technique that identifies the presence of secondary ice production (SIP) in supercooled mixed-phase clouds (Luke et al., 2021). $V_{AIR}$ is estimated from the radar Doppler spectra using the location (in m s$^{-1}$) of the slower falling edge of the supercooled liquid spectral density's principal peak and is adjusted by a value of 0.28 m s$^{-1}$ to compensate for turbulence broadening. The selected velocity adjustment for turbulence broadening of the radar Doppler spectra is applicable only to radars operating with similar characteristics to the ARM KAZRs. The value of the "climatological correction" is based on multi-year analysis of KAZR observations in mixed-phase and liquid clouds (Luke et al., 2021; Zhu et al., 2022). Our primary measurements in this analysis are linear depolarization ratio (LDR) determined by the ratio of the cross-polarized to co-polarized spectral density, calibrated co-polarized spectral reflectivity normalized to units of dBZ / m s$^{-1}$, and spectral terminal fall speed computed as the difference between $V_{AIR}$ and spectral $V_D$.

**Comment 9**: In Lines 191-192, when clouds only extend over 0.5-3 km (this was mentioned earlier), how many 20-minutes samples could be derived given these short spatial scales?

**Response**: For clouds with durations less than 20 minutes, we use all the cloud profiles to retrieve EDR. The number of the retrieved EDR for clouds with spatial scales of 0.5 – 3 km depends on the horizontal wind speed. For example, if the wind speed is 5 ms$^{-1}$, a cloud with a horizontal scale of 2 km corresponds to roughly 7 minutes of radar observations. In this case, all 7 minutes of observations are used for the retrieval, and one EDR product will be retrieved for this cloud. We thank the reviewer for pointing this out and have added a description to the manuscript.

**Change**: For clouds with durations less than 20 minutes, Doppler velocities for all the collected cloud profiles are used to generate S(f) …

**Comment 10**: In Lines 212-213, which 'typical' studies used the 'typical' value of 2 m s$^{-1}$ to discriminate stratiform from convective situations?

**Response**: We will add references for typical vertical velocity values in stratus and stratocumulus.

**Change**: In addition, the 2 m s$^{-1}$ $V_{AIR}$ threshold ensures we exceed the typical vertical air motion values observed in stratus and stratocumulus (Guibert et al., 2003; Peng et al., 2005; Guo et al., 2008; Ghate et al., 2010; Hudson and Noble, 2014).

**Added Citations**:
Ghate, V.P., Albrecht, B.A., and Kollias, P.: Vertical velocity structure of nonprecipitating continental boundary layer stratocumulus clouds, Journal of Geophysical Research: Atmospheres, 115, https://doi.org/10.1029/209JD013091, 2010

Guibert, S., Snider, J.R., and Brenguier, J.: Aerosol activation in marine stratocumulus clouds: 1. Measurement validation for a closure study, Journal of Geophysical Research: Atmospheres, 108, https://doi.org/10.1029/2002JD002678, 2003.

Guo, H., Liu, Y., Daum, P.H., Senum, G.I., and Tao, W.: Characteristics of vertical velocity in marine stratocumulus: comparison of large eddy simulations with observations, Environmental Research Letters, 3, https://doi.org/10.1088/1748-9326/3/4/045020, 2008.

Hudson, J.G, and Noble, S.: CCN and Vertical Velocity Influences on Droplet Concentrations and Supersaturations in Clean and Polluted Stratus Clouds, Journal of the Atmospheric Sciences, 71, 312-331, https://doi.org/10.1175/JAS-D-13-086.1, 2014.

Peng, Y., Lohmann, U., and Leaitch, R.: Importance of vertical velocity variations in the cloud droplet nucleation process of marine stratus clouds, Journal of Geophysical Research: Atmospheres, 110, https://doi.org/10.1029/2004JD004922, 2005.

**Comment 11**: In Lines 220-222, what makes the authors assume that the updraft regions travel with the horizontal wind? Can they rule out the presence of standing/rolling waves? The frequent appearance of cloud straits in cold-air outbreaks make we wonder whether the clouds travel parallel to the wind vector or if they are trapped in a roll-over vortex. Satellite images might help to demonstrate the validity of the assumption of the authors.

**Response**: We agree that it is difficult to understand how atmospheric variables in horizontal space impact measurements from vertically-pointing instruments. We assumed the regions travel with the horizontal wind and that we could rule out rolls because our updrafts are vertically oriented and do not display any sort of tilt.

**Comment 12**: In Line 232, how was cloud top derived? Are there uncertainties related to the derived cloud height values?

**Response**: We will make this clearer in the paper. We simply looked at the heights at which we saw the last echo in the KAZR field during this period and gave the temperature range at those heights from the nearest sounding in time.

**Change**: Meanwhile, temperatures ranged from -44.6 °C to -37.5 °C near cloud top, where we took the cloud tops to be the last detectable echoes in the KAZR columns.

**Comment 13**: In Line 233, to which height region does the statement about wind shear apply? Or was it the same in the whole troposphere (or at least at all height levels)? This is rather unlikely in convective situations.

**Response**: This extends to the top of the cloud layer you see in Figure 2. We will make this clearer.

**Change**: Within the region from the surface to the range of cloud tops, the lapse rate was about 8.4 °C km$^{-1}$, and the prevailing wind was predominantly from the northwest with at most 8 – 9° of wind shear.

**Comment 14**: In Line 237, how can a cloud with top temperatures of below -40°C show no ice formation? Could it be that the cloud was snow-dominated, but melting was not detected due to the cold cloud base and low (below 0°C) dew point? In general, dew point needs to exceed 0°C to trigger melting of the particles. This is, e.g., the general assumption in the ACTRIS Cloudnet algorithms.

**Response**: The reviewer is correct. The selected CAO cases show no evidence of the melting process. No radar bright band is observed, and no increase in the Doppler velocity is observed. The CAO cumulus clouds are composed of solid hydrometeors, except the areas with strong updraft motions where the presence of supercooled liquid is possible considering the large LWP values recorded by the MWR.

**Comment 15**: In Lines 244-245, can Mie-scattering conditions be excluded? Or could it be that large cloud droplets or rain produced artificially high brightness temperatures and consequently overestimated values of LWP?

**Response**: The two-channel MWR used in ARM uses wavelengths of 23.4 and 31.8 GHz. The use of low frequency channels greatly reduces the impact of Mie-scattering on the observed brightness temperatures.

**Comment 16**: In Lines 268-273, really nice approach! Just wanted to point this out.

**Response**: Thank you very much for your positive comment!

**Comment 17**: In Lines 299-302, why does $V_{AIR}$ drop for high LWP values? Mie effects? Can Mie effects be negated?

**Response**: We have made changes to Figure 7 and have ignored the high LWP values which occur infrequently. The very small sample size is responsible for the apparent drop in $V_{AIR}$. We will adjust Figure 7 and make the appropriate changes to Section 4.3.

**Change**: Visual inspection of Figure 2 suggests a correlation between the presence of liquid water and the coherent updraft structures. This relationship is further investigated using all the COMBLE observations (Fig. 7). Looking at the LWP measurements broadly, Figure 7a shows nearly 75% of the LWP data is near zero, while only about 1% of the LWP data is higher than 2 kg m$^{-2}$. Meanwhile, Figure 7b shows that as LWP in the column increases, so too does the maximum $V_D$, which reinforces the visual inspections we made and our use of LWP as a threshold for our vertical air motion retrieval. In general, the LWP correlates well with the square of the depth of the cloud (Wood, 2012; Fan et al., 2018). In Figure 7c, the measured LWP is plotted against the sum of $V_{AIR}$ values in the updraft depth. Similarly, in Figure 7d, the LWP in the column is plotted against the maximum $V_{AIR}$ value in the updraft depth. These two

relationships exhibit a plausible agreement between two independent measurements, which further supports the good performance of the $V_{AIR}$ retrieval technique.

[Figure]

**Figure 7. For all 13 COMBLE cases, (a) the relative frequency of liquid water path (LWP) in bins of width 0.1 kg m⁻², (b) the median, 25th, and 75th percentile of the maximum Doppler velocity ($V_D$) in the atmospheric column for each LWP bin of width 0.25 kg m⁻² or 0.5 kg m⁻², (c) the median, 25th, and 75th percentile of the sum of vertical air velocity ($V_{AIR}$) in the updraft depth for each LWP bin of width 0.25 or 0.5 kg m⁻², and (d) the median, 25th, and 75th percentile of the maximum $V_{AIR}$ in the updraft depth for each LWP bin of width 0.25 or 0.5 kg m⁻².**

**Comment 18**: In Figure 1, what is the saturation region of KAZR? I.e., what is the upper limit of detectable reflectivity? How would any saturation effects of the cloud radar affect the presented retrieval?

**Response**: Excellent comment. Saturation of millimeter-wavelength radars is common in the presence of strong echoes near the radar (within the first 1-2 km). All the analysis presented in this manuscript has been conducted using the KAZR general mode, which operates a short pulse. The use of a short pulse results in a higher value of reflectivity where saturation occurs. Receiver saturation depends on height. The figure below provides the maximum reflectivity recorded by the KAZR at each range gate. As we can see, the KAZR radar reflectivity continues to grow or at least is not decreasing as we get closer to the radar. This suggests that even the highest radar reflectivities recorded by the KAZR did not saturate the receiver. This was one of the changes introduced to the ARM KAZRs more than a decade ago (Kollias et al., 2020).

[Figure]

Minor comments:

**Comment 19**: In Lines 34-36, I suggest providing some quantitative information, i.e. how large were the eddy dissipation rates? How was the evidence of secondary ice formation derived? How intense was it during COMBLE?

**Response**: We accept this suggestion and will add quantitative information about the eddy dissipation rates and brief background information about how the secondary ice production evidence was derived. However, we cannot make any definitive statements about COMBLE, as we only show data from one of our 13 cases.

**Change**: The CAO cumulus clouds exhibit values between $10^{-3}$ and $10^{-2}$ m$^2$ s$^{-3}$ in the lowest 2 km of the atmosphere, and using a radar Doppler spectra technique, evidence of secondary ice production is found during one of the cases.

**Comment 20**: In Lines 97-102, KAZRs provide usually General Mode and Sensitive Mode. Which one was used? Or even better - link to the utilized datasets in the ARM database. Same holds for the other instruments.

**Response**: We will make note of this in the paper; we used the general mode of KAZR. All the COMBLE-specific DOIs for each instrument are listed in the Data Availability section.

**Change**: In this study, the radar reflectivity factor and mean Doppler velocity from the general mode are used.

**Comment 21**: In Lines 104-106, how often were sondes launched?

**Response**: The sondes are launched every six hours, where the maximum number of sondes per day would be 4 (~0600 UTC, ~1200 UTC, ~1800 UTC and ~0000 UTC). We will make note of this.

**Change**: The balloon-borne sounding system (SONDE), in which soundings are launched every 6 hours, and the Interpolated Sonde (INTERPSONDE) value-added product…

**Comment 22**: In Line 109, further questions arise about the ceilometer dataset: (1) which of the three cloud bases was selected? (2) The doi links to many different CEIL datasets. It is possible to only link to the COMBLE dataset? (3) which type of ceilometer was operated?

**Response**: We will make note of this in the paper; we used the first or lowest cloud base height detected. Regarding the DOI, if you click on it, you are given general information about the ceilometer on the ARM archive, but the DOI listed is the COMBLE-specific DOI. Finally, ARM used a Vaisala Laser Ceilometer, which we will make note of.

**Change**: We use a Vaisala laser ceilometer, which sends a laser pulse at a 910 nm wavelength to detect light scattered by clouds and precipitation, to retrieve the lowest or first cloud base height (Morris, 2016).